# Regular testing of asymptomatic healthcare workers identifies cost-efficient SARS-CoV-2 preventive measures

Daniel Sanchez-Taltavull[1,2,3]*, Violeta Castelo-Szekely[1,2,3], Shaira Murugan[1,2,3], Jonathan I. D. Hamley[1,2,3], Tim Rollenske[1,2], Stephanie C. Ganal-Vonarburg[1,2], Isabel Büchi[1,2,3], Adrian Keogh[1,2,3], Hai Li[1,2], Lilian Salm[1,2,3], Daniel Spari[1,2,3], Bahtiyar Yilmaz[1,2], Jakob Zimmermann[1,2], Michael Gerfin[4], Edgar Roldan[5], Guido Beldi[1,2,3]*, UVCM-COVID researchers[1,2,3]¶

**1** Department of Visceral Surgery and Medicine, Bern University Hospital, University of Bern, Bern, Switzerland, **2** Department of Biomedical Research, University of Bern, Bern, Switzerland, **3** Bern Center for Precision Medicine, Bern, Switzerland, **4** Department of Economics, University of Bern, Bern, Switzerland, **5** ICTP, The Abdus Salam International Centre for Theoretical Physics, Trieste, Italy

¶ Membership of the UVCM-COVID researchers is listed in the Acknowledgments.
\* daniel.sanchez@dbmr.unibe.ch (DST); guido.beldi@insel.ch (GB)

**Data Availability Statement:** All relevant data and algorithm are within the paper and its Supporting information files.

## Abstract

Protecting healthcare professionals is crucial in maintaining a functioning healthcare system. The risk of infection and optimal preventive strategies for healthcare workers during the COVID-19 pandemic remain poorly understood. Here we report the results of a cohort study that included pre- and asymptomatic healthcare workers. A weekly testing regime has been performed in this cohort since the beginning of the COVID-19 pandemic to identify infected healthcare workers. Based on these observations we have developed a mathematical model of SARS-CoV-2 transmission that integrates the sources of infection from inside and outside the hospital. The data were used to study how regular testing and a desynchronisation protocol are effective in preventing transmission of COVID-19 infection at work, and compared both strategies in terms of workforce availability and cost-effectiveness. We showed that case incidence among healthcare workers is higher than would be explained solely by community infection. Furthermore, while testing and desynchronisation protocols are both effective in preventing nosocomial transmission, regular testing maintains work productivity with implementation costs.

## Introduction

During the outbreak and spread of an epidemic such as that of SARS-CoV-2, protection of individuals relies on non-pharmaceutical interventions (NPIs) at least until a large fraction of the population is vaccinated. Common NPIs include wearing protective equipment, social distancing and working from home, as well as isolation of symptomatic individuals. However, applying some of these measures among healthcare workers is challenging, as they need to work in close contact with infected patients and with their coworkers, and as the possibilities

**Funding:** The study was supported by the Swiss National Science Foundation with the grants 196059 (G. B., D. S.-T.) and 196641 (S. G.-V.) and by the Bern Center for Precision Medicine (G. B., D. S.-T.). The funders had no role in study design, data collection and analysis, decision to publish, or preparation of the manuscript.

**Competing interests:** The authors have declared that no competing interests exist.

for home-office options are limited. Therefore, the healthcare workforce could be exposed to an increased risk of infection transmission than the general population and devising protective strategies for them remains a crucial task. In addition to the above, other NPIs have been considered through modelling and case studies, notably, testing of healthcare personnel and implementation of desynchronisation to reduce worker contact.

Studies of testing healthcare workers have shown differing results with regard to whether they have a reduced or increased case incidence with respect to the general population. The incidence among health workers seems to be increased [1] although it mirrored the curve in the general population [2]. Other studies showed no major occupational risk for healthcare professionals [3, 4]. Together, these results indicate that the exposure risk among healthcare workers likely depends on the specificities of the study as well as on the local magnitude of the epidemic, the availability and implementation of basic protective measures and the effective adherence to them in the studied scenario. As a caveat, most of these studies carried out testing only on symptomatic individuals and did not assess whether widespread regular testing of asymptomatic and pre-symptomatic coworkers reveals a higher incidence and can prevent transmission at the workplace in an epidemic in which a significant proportion of affected individuals can present without symptoms [5, 6].

The implementation of desynchronising the presence of workers at the workplace has been modelled in the context of the SARS-CoV-2 epidemic. A model of 4-day work, 10-day lockdown cycle showed that these cycles can allow economic activity to partially continue by exploiting the latent period of the virus and allowing those infected at work to reach maximum infectiousness while at home [7]. Similarly, organisation of laboratory or clinical staff into smaller groups working shorter shifts can lead to reduced interaction and transmission among coworkers [8, 9]. Yet such models have not been tested in with experimental data during the current pandemic.

In this study we report the results of a testing pipeline of one clinical department that has been introduced from the beginning of the epidemic in Switzerland. Starting from March 23rd 2020, we implemented a RT-PCR-based testing pipeline that was applied weekly on a voluntary basis to a cohort of asymptomatic healthcare workers. Based on the results of this testing pipeline we have developed a mathematical model that includes the transmission dynamics of SARS-CoV-2 in the Swiss Canton of Bern, and that integrates the infection forces among a group of healthcare workers—including the general population, hospitalised patients and coworker—as independent sources of infection. Individuals testing positive were referred for confirmation to a second testing pipeline carried out by the official authorities, and sent into self-isolation.

We used this data not only to detect and isolate pre- and asymptomatic cases, but also to inform the model on the contribution of worker-worker and worker-patient contact to case incidence among the studied healthcare group. As a comparison, we have modelled a work desynchronisation strategy in which healthcare workers would shift weekly between in-hospital and at-home work, as a way to contain infection transmission by reducing contact between colleagues.

Given that the number of people tested may underestimate the true incidence, we have estimated the true incidence based on the number of hospitalized patients as it is a more reliable measure. Therefore, we fitted the model to the recorded number of hospitalised individuals in the canton of Bern to infer the time-varying infection rate of the general population, and we estimated the additional infection forces affecting the health workforce by fitting the individuals testing positive and sent to quarantine. The model allowed us to study how both regular testing of all individuals and desynchronisation strategies are efficient in preventing workers from infecting each other under the different scenarios, and we present how the model can

help decision making on whether to implement such strategies. Furthermore, we studied the impact that the implementation of these interventions would have on productivity and the availability of the workforce, and compare them from an economic point view.

## Materials and methods

### Epidemiological data

We collected publicly available data on patient hospitalizations, deaths and confirmed cases in the Canton of Bern from March 1st 2020 to March 1st 2021, from the BAG (Bundesamt für Gesundheit) [10]. From the publicly available data we selected the sum of the last 7 days of infected and hospitalizations in the region of Bern.

### Internal testing pipeline

The Study was approved by the ethical committee of the Canton of Bern (2020–00563). A cohort study was performed that included healthcare workers of the Department for Visceral Surgery and Medicine of the Bern University Hospital. A testing protocol was established and implemented weekly on a voluntary basis, starting March 23, 2020. Informed consent was obtained by all participants. Consenting participants could get tested either on Monday or Thursday every week, and positive cases were sent for a confirmatory test carried out by official institutions and isolated.

**Sample collection.**    Healthcare workers collecting samples from the participants were instructed to use personal protective equipment including FFP-2 face masks, eye protection and face shield, two layers of gloves and two layers of hairnets. Nasopharyngeal specimens were obtained by slowly and gently inserting a MiniTip swab through the nostril palate and closely to the septum until contact with the nasopharynx was reached. The swab was removed after gently rubbing and rolling at the nasopharynx. Rarely, oral swabs were taken instead by taking samples from the posterior oropharynx. Specimens taken from each participant were immediately placed into 2-ml tubes containing TRIzol (Invitrogen), an acid-guanidinium-phenol based reagent that inactivates the virus and stabilises the RNA. All collected samples were sprayed with 70% EtOH before leaving the collection site. Specimens were further processed on the same day or kept at -20˚C until processed the following day.

**RNA extraction and RT-qPCR.**    Laboratory workers handling the collected samples were instructed to use personal protective equipment including gowns, facemask and two layers of gloves. The virus was inactivated by vortexing the TRIzol tube containing the swab-tips and 3.3 ng carrier RNA (Qiagen) was added. Then 200 $\mu$l of chloroform was added to each sample and the tubes were inverted multiple times, followed by centrifugation for 10 min at 13,200 rpm. RNA purification and concentration was carried out using RNA easy-mini elute kit (Qiagen) according to the manufacturer's protocol. Briefly, 270 $\mu$l of the clear aqueous phase of each sample was re-suspended with 945 $\mu$l of RLT buffer and 670 $\mu$l of 100% EtOH. The sample was added to the column in three steps with intermittent centrifugation at 8000 rpm for 15 s allowing all sample to pass through the column. 500 $\mu$l of RPE buffer was added to each column as a first washing step and centrifuged at 8000 rpm for 15 s. Subsequently, 500 $\mu$l of 80% EtOH was added to each column and centrifuged at 13,200 rpm for 1 min. After drying the RNA containing membrane by centrifugation at 13,200 rpm for 3 min, 20 $\mu$l RNAse-free water was added to elute the RNA. The sample was incubated for 1 min at room temperature before it was centrifuged for 1 min at 13200 rpm.

One-step combined reverse transcription and qPCR using SuperScript III One-Step RT-PCR System with Platinum Taq DNA Polymerase (Invitrogen) was performed using the manufacturers recommended protocol. Briefly, 2 $\mu$l of each RNA sample was amplified using

primers for SARS-Cov2 E-gene, which codes for protein forming pores on the host membrane, and human RNase-P, in order to verify successful sampling of the healthcare workers. Technical triplicates were performed. The PCR was run under following conditions: 50˚C for 10 min, 95˚C for 3 min, followed by 45 cycles of 95˚C for 15 s and 58˚C for 30 s on the BioRad CFX 384 real time PCR system.

The results for the RT-qPCR are displayed by ct (cycle threshold) values. In the current study the detection limit is at a ct of 40 and above. Values below a ct of 32 indicate high viral titers.

## Mathematical modelling

**Transmission dynamics in the city model.**   We developed a population-based model that integrates the transmission dynamics in the Swiss Canton of Bern as a deterministic model, and the infection dynamics among a group of healthcare workers, studied as a stochastic system (Fig 2A). The Canton dynamics are given by the following system of differential ordinary equations (ODE):

$$\frac{dS_C}{dt} = -\alpha(t)S_C I_C \tag{1}$$

$$\frac{dE_C}{dt} = \alpha(t)S_C I_C - sE_C \tag{2}$$

$$\frac{dI_C}{dt} = sE_C - r(1 - \epsilon_1)I_C - h\epsilon_1 I_C \tag{3}$$

$$\frac{dH_C}{dt} = h\epsilon_1 I_C - w(1 - \epsilon_4)H_C - d\epsilon_4 H_C \tag{4}$$

$$\frac{dR_C}{dt} = r(1 - \epsilon_1)I_C + w(1 - \epsilon_4)H_C \tag{5}$$

$$\frac{dD_C}{dt} = d\epsilon_4 H_C \tag{6}$$

Susceptible individuals in the Canton, $S_C$, get exposed at a rate $\alpha$; exposed individuals, $E_C$, then become infectious, $I_C$, after a latency period $s^{-1}$. A proportion $\epsilon_1$ of infected individuals are hospitalised, $H_C$, at a rate $h$. Infectious individuals recover, $R_C$ at a rate $r$, whereas hospitalised patients recover at a rate $w$. We only considered deaths from hospitalised people: a proportion $\epsilon_4$ of $H_C$ die at a rate $d$. $N$ refers to the total population in the Canton of Bern.

As in Althaus et al. [11], the transmission rate $\alpha$ captures the reduction in the infections following the implementation of NPIs by the government, described as:

$$\alpha(t) = \frac{\pi}{N}\left(1 - \frac{1 - \kappa}{1 + e^{-v(t-\tau)}}\right) \tag{7}$$

where $\frac{\pi}{N}$ is the infection at the start of the epidemic, $\kappa$ is the relative transmission reached after implementation of NPIs, $v$ is the slope of the sigmoid function, and $\tau$ is the midpoint of transmission reduction.

**Transmission dynamics among the healthcare workers model.** The transmission dynamics among the HCW are formulated in terms of the Master Equation [12]

$$\frac{\partial P(\mathbf{X}, t)}{\partial t} = \sum_j \left( W_j(\mathbf{X} - r_j, t) P(X - r_j, t) \right) - W_j(\mathbf{X}, t) P(\mathbf{X}, t)) \tag{8}$$

where $X = (S, E, I, A, R, Q)$ is the state vector containing the number of healthy workers susceptible to infection, $S$, the number of exposed workers who are infected but still not contagious that are working, $E$, the number of symptomatic infected workers, $I$, the number of asymptomatic infected workers, $A$, that are working, the number of recovered workers $R$, and the number of exposed and asymptomatic workers that have been identified as infected with a PCR test and put in quarantine, $Q$. $P(X, t)$ is the probability of the population of having value $X$ at time $t$. The transition rate per unit time of the process $j$ is given by $W_j(X, t)$, and $r_j$ is the state change in $X$ upon the occurrence of the process $j$. All reactions and rates are described in Table 1. Due to the small size of our cohort, we do not consider differences such as age or type of job in the model.

The mean-field dynamics of the stochastic system are given by the following system of ODEs:

$$\frac{dS}{dt} = -\alpha(t)aSI_C - \beta S \frac{A}{N_H} - \gamma S \frac{H_C}{N} \tag{9}$$

$$\frac{dE}{dt} = \alpha(t)aSI_C + \beta S \frac{A}{N_H} + \gamma S \frac{H_C}{N} - s\epsilon_2 E - s(1 - \epsilon_2)E - p(t)\epsilon_3 E \tag{10}$$

$$\frac{dI}{dt} = s\epsilon_2 E - rI \tag{11}$$

$$\frac{dA}{dt} = s(1 - \epsilon_2)E - rA - p(t)\epsilon_3 A \tag{12}$$

$$\frac{dR}{dt} = rA + rI + rQ \tag{13}$$

$$\frac{dQ}{dt} = p(t)\epsilon_3 E + p(t)\epsilon_3 A - rQ \tag{14}$$

**Table 1. Reaction rates of the stochastic dynamics described by Eq 8.**

| Transition rate | State change vector $r_j = (\Delta S, \Delta E, \Delta I, \Delta A, \Delta R, \Delta Q)$ | Description |
|---|---|---|
| $W_1 = \alpha(t)aI_C S$ | (−1, 1, 0, 0, 0, 0) | $S \rightarrow E$ |
| $W_2 = \frac{\beta}{N_H} AS$ | (−1, 1, 0, 0, 0, 0) | $S + A \rightarrow E + A$ |
| $W_3 = \frac{\gamma}{N} SH_C$ | (−1, 1, 0, 0, 0, 0) | $S \rightarrow E$ |
| $W_4 = sE\epsilon_2$ | (0, −1, 1, 0, 0, 0) | $E \rightarrow I$ |
| $W_5 = sE(1 - \epsilon_2)$ | (0, −1, 0, 1, 0, 0) | $E \rightarrow A$ |
| $W_6 = rI$ | (0, 0, −1, 0, 1, 0) | $I \rightarrow R$ |
| $W_7 = rA$ | (0, 0, 0, −1, 1, 0) | $A \rightarrow R$ |
| $W_8 = p(t)\epsilon_3 E$ | (0, −1, 0, 0, 0, 1) | $E \rightarrow Q$ |
| $W_9 = p(t)\epsilon_3 A$ | (0, 0, 0, −1, 0, 1) | $A \rightarrow Q$ |
| $W_{10} = rQ$ | (0, 0, 0, 0, 1, −1) | $Q \rightarrow R$ |

Susceptible workers, $S$, become exposed at a rate $\alpha$ due to contact with the general population ($I_C$), at a rate $\beta$ from being in contact with an asymptomatic coworker ($A$, see later), and at a rate $\gamma$ from contact with infected hospitalised individuals ($H_C$). Exposed individuals ($E$) then develop the disease after a latency period $s^{-1}$, and become either infectious and symptomatic, $I$, or infectious and asymptomatic, $A$, with the probability of being $I$ or $A$ given by $\epsilon_2$ and $1-\epsilon_2$ respectively. Exposed and asymptomatic can be identified by testing, where $\epsilon_3$ is the probability of obtaining a true positive in the test, and the details of the testing are described below. For simplicity, we write $p(t)$ in the equations to show it is time dependent. Finally, $r$ is the recovery rate.

Both asymptomatic and symptomatic individuals recover at a rate $r$. We did not considered deaths among healthcare workers. $N_H$ refers to the number of healthcare workers participating in the study. Due to their small number, their transmission dynamics are studied as a stochastic process using Gillespie simulations [13].

**Modelling a desynchronisation strategy.**  We modelled a desynchronisation strategy in which the workforce was split into two non-mixing groups to establish a weekly rotation shift: Group 1 works in-hospital during odd weeks and at-home during even weeks, and Group 2 works at-home during odd weeks and in-hospital during even weeks.

To mathematically formulate the desynchronisation, each variable is dichotomised in two non-interacting groups ($S_1$, $E_1$, $I_1$, $A_1$, $R_1$, $Q_1$) and ($S_2$, $E_2$, $I_2$, $A_2$, $R_2$, $Q_2$), each of them described by Eq 8. When workers are doing home office, they cannot be infected by other co-workers nor by patients. That is, the parameters $\beta$ and $\gamma$ become time dependent, being $\beta_1(t) = 1.5\beta$, $\gamma_1(t) = 1.5\gamma$, $\beta_2(t) = 0$, $\gamma_2(t) = 0$ during the odd weeks, and exchanging values during the even weeks, $\beta_1(t) = 0$, $\gamma_1(t) = 0$, $\beta_2(t) = 1.5\beta$, $\gamma_2(t) = 1.5\gamma$. We assume an increase of 50% in $\beta$ and $\gamma$ because some tasks imply cooperation between workers.

**Modelling testing.**  We modelled the testing as an instantaneous reaction in which $E \rightarrow Q$ or $A \rightarrow Q$. Every $P$ days, for each infected worker we generate a random number, $x$ in (0, 1), if $x < \epsilon_3 f_p$, where $f_p$ is the fraction of workers tested and $\epsilon_3$ is the probability of detecting a true positive, the worker is sent to quarantine. During the first wave, that is $t < 91$, on average 65% of the workers were tested every week, afterwards 45% were tested every week. Thus, $f_p = 0.65$ for $t < 91$ and $f_p = 0.45$ for $t \geq 91$.

**Model fitting.**  *City*. In Switzerland the first lockdown was started on March 16th 2020 that included reduction of elective healthcare-associated interventions such as surgery until April 27th 2020. With the onset of the second wave national measures were again installed by October 19th 2020. After a second peak in 30th October 2020 the second wave faded by end of January 2021.

$\tau$, $\pi$, $\nu$ and $\kappa$ were chosen to obtain the best fit of the model to the data, by minimizing

$$G(\kappa, \nu, \pi, \tau) = \sum_{i=0}^{n} ((H_c^i(t_i) - O(t_i))^2, \tag{15}$$

with $t_i \in \{0, 7, 14, \ldots\}$, $O(t_i)$ the sum of the number of cases of the last 7 days at time $t_i$ starting March 1st 2020 ($t = 0$) and

$$H_c^i(t_i) = \int_{t_i-7}^{t_i} h\epsilon_1 I_c dt$$

The fitting was done in 4 parts:

1. Wave 1: With the initial condition ($S_C$, $E_C$, $I_C$, $H_C$, $R_C$, $D_C$) = ($N - 1$, 0, 1, 0, 0, 0) starting Day -8 to Day 91, with $\alpha(t)$ described by 7 resulting into $\tau = 17.4352$, $\pi = 1.2801$, $\kappa = 0.016$,

$v$ = 1.5. The optimization was performed with the R function *optim* with *method="L-BFGS-B"* in the parameter region $\pi \in [0, 2]$, $v \in [0, 1.5]$, $\kappa \in [0, 0.15]$, and $\tau \in [15, 24]$.

2. Summer: Day 91 to Day 189
In this scenario, we consider a different $\alpha(t)$,

$$\alpha(t) = \frac{\pi}{N} \tag{16}$$

The reason for this difference, is that during this period, there were no remarkable non-therapeutic interventions to reduce the number of cases. Instead, we assume it to be a period where the use of contact-tracing measures avoided a large increase in the number of cases. This difference is equivalent to the assumption $\kappa = 1$.
The optimization resulting into $\pi = 0.2869$. The optimization was performed with the R function *optim* with *method="L-BFGS-B"* in the parameter region $\pi \in [0, 1]$.
Eventually, the contact-tracing measures were overwhelmed by the number of cases, and the second wave started, were we considered a Holling type-II function to describe the dynamics.

3. Wave 2: Day 189 to Day 273, with $\alpha(t)$ described by 7 resulting into $\tau = 240.2609$, $\pi = 0.3177$, $\kappa = 0.44$, $v = 0.8934$. The optimization was performed with the R function *optim* with *method="L-BFGS-B"* in the parameter region $\pi \in [0, 0.7]$, $v \in [0, 1.5]$, $\kappa \in [0, 1]$, and $\tau \in [230, 246]$.

4. Winter: Day 273 to Day 365, with $\alpha(t)$ described by 7 resulting into $\tau = 301.6273$, $\pi = 0.1919$, $\kappa = 0.6048$, $v = 1.5$. The optimization was performed with the R function *optim* with *method="L-BFGS-B"* in the parameter region $\pi \in [0, 0.5]$, $v \in [0, 1.5]$, $\kappa \in [0, 1]$, and $\tau \in [295, 305]$.

The trajectory of $\alpha$ is shown in S1 Fig. The selection of the range for $v$ is arbitrary. We limited it to 1.5 because it would produce a change in $\alpha$ faster than the temporal resolution of our data and we decided not to allow this. In S2 Fig we show the fit without this limitation and the difference is negligible.

*Hospital.* In order to protect workers, the department started a desynchronisation strategy on March 16th 2020 and ended on May 4th 2020. Moreover, the department implemented a weekly testing regime starting March 23rd 2020. During the desynchronisation period, only the in-hospital group were tested, on Monday and Thursday. For simplicity we assume all workers, in-hospital or doing home office, were tested once a week and the same day. This assumption, is equivalent to assuming the second test was delayed until Monday.

In order to mimic the hospital dynamics, we simulated our system as follows.

- Set initial conditions to $S_C = N - 1$, $I_C = 1$, $E_C = H_C = D_C = R_C = 0$ and $t = -8$ and integrate the ODE systems 1–6 until $t = 15$. No that March 1st 2020 corresponds to $t = 0$.

- set initial condition to $S_1 = S_2 = 157$ and start stochastic simulation

  1. Select reaction $r_i$ and $\Delta t$ according to the Gillespie algorithm.

  2. Update the variables.

  3. Integrate the ODE system for $\Delta t$.

  4. Starting $t = 22$ we perform tests weekly.

  5. At $t = 64$ the desynchronisation stops and the two groups are merged together.

6. If $t < t_{end}$ go to 1. Otherwise end the simulation.

All details of the simulation algorithm have been included in S1 File.

To avoid large step-sizes selected by the Gillespie algorithm, we added an artificial reaction $\emptyset \to \emptyset$ at a rate 100 days$^{-1}$. This reaction avoids possible problems coupling the deterministic variables of the city with the stochastic variables of the hospital. For example, when $\alpha(t)$ is small and all workers are susceptible or recovered, the Gillespie algorithm would select a large step size. However, $\alpha$ could increase substantially during that long period of time, leading to possible errors.

The fit for the first wave was done by minimizing the equation

$$G(\beta, \gamma) = \sum_{i=0}^{n} ((Q^i(t_i) - O(t_i))^2, \tag{17}$$

at times $t_i = 61$ and $t_i = 91$, where $Q^i(t)$ is the cumulative $Q$ at time $t$ and $O(t)$ is the cumulative positive tests in the experiment. The two selected time points represent the end of the first wave, selected to account for the total number of infected, and a month after a period in which there were no detected cases.

Analogously, the fit for the second wave was done by minimizing Eq 17 at $t_i = 191$, 291, and 346.

*Work productivity*. In order to study how protective strategies affect work output we defined a variable, $W$, accounting for work done by healthy individuals, and given by:

$$\frac{dW}{dt} = S + E + A + R + \epsilon_5 \mu Q \tag{18}$$

where $\epsilon_5$ indicates the proportion of individuals in quarantine that feel healthy and can work (i.e. asymptomatic) and $\mu$ is the productivity of a health worker when in quarantine/working from home compared to that when working at the hospital. In particular, $\mu$ accounts for the fact that the tasks that can be carried out from home by most healthcare workers are limited and will thus reduce their overall normal output [14].

In the case of individuals in a desynchronisation strategy, the work done is given by:

$$\frac{dW}{dt} = (S_1 + E_1 + A_1 + R_1 + \epsilon_5 \mu Q_1) + \mu(S_2 + E_2 + A_2 + R_2 + \epsilon_5 Q_2) \tag{19}$$

In our simulations, the worked is accounted between $t = 15$ and $t = 364$.

*Parameters of the model*. All the parameters of the model are given in Table 2.

## Cost analyses

In order to compare the intervention strategies from an economic point of view, we estimated the cost of both testing and desynchronisation. We estimated the cost of each test to be $\sim$ USD 42 which includes the cost of swabbing material, protective masks, and extraction and qPCR reagents, and the salary of the workers doing the tests. The cost of testing is described by

$$c = (\text{cost per test}) \cdot (\text{Number of tests})$$

The cost of home-office during a desynchronisation strategy was considered to be the proportion of the monthly salary that would correspond to the unproductive period. The average monthly wage of a healthcare professional in Switzerland was obtained from the International Labour Organisation for year 2016 (USD 7065.2) [18].

## Data availability

The data of the internal testing pipeline is included as S2 File.

## Results

### Implementation of an internal RT-PCR-based testing protocol on healthcare worker

Starting March 23, 2020, we implemented an RT-PCR-based testing pipeline as a clinical service of the department for Visceral Surgery and Medicine at the Bern University Hospital (Fig 1A). The RT-PCR was developed within our department. Consenting participants were tested in a voluntary basis in one of the two rounds of testing performed weekly, regardless of whether they presented symptoms. Our population consists of 314 persons working in contact with patients and 58 persons who were not working in contact with patients who all attended at least five sessions or were identified positive.

Throughout the study period, the number of weekly tests performed among the 314 healthcare workers in contact with patients is shown in Fig 1A, corresponding to a weekly participation of $\sim$ 65% in the first wave and a participation of $\sim$ 45% during the second wave and until end of February. Included individuals were aged 18–64 and working in different healthcare roles, including medical doctors, nurses, medico-technical and laboratory staff, in potential direct contact with COVID-19 patients or samples.

At the end of the reported study and considering an RT-PCR Ct cut-off of 32 (see Methods), 15 individuals in potential direct contact with COVID-19 patients or samples tested positive (Fig 1A, red bars), and with a cut-off of 40, positive tests were observed in 30 individuals.

Positive cases were informed, were retested by the official testing protocol, and sent to self-isolation even if asymptomatic.

**Table 2. Parameters of the models.**

| Parameter | Value | Description |
|-----------|-------|-------------|
| $a$ | varied (see text) | fraction of transmission rate outside the hospital |
| $\beta$ | varied (see text) | Transmission rate between coworkers |
| $\gamma$ | varied (see text) | Transmission rate from hospitalised patients |
| $1/s$ | 5.2 days [15] | Latency period |
| $1/r$ | 6 days [16] | Recovery rate |
| $1/w$ | 10 days [16] | Recovery rate for hospitalised patients |
| $1/h$ | 6 days [16] | Hospitalisation rate |
| $1/d$ | 10 days | Death rate |
| $\epsilon_1$ | 2.5% | Probability of being hospitalised |
| $\epsilon_2$ | 30% (assumption) | Probability of being Infectious symptomatic |
| $\epsilon_3$ | 95% | Probability of a true positive test |
| $\epsilon_4$ | 0.37 | Probability of death |
| $\epsilon_5$ | 0.7 | Proportion of healthy individuals in quarantine |
| $f_p$ | varied (see text) | Fraction of workers tested |
| $\pi$ | estimated | infection rate in the city |
| $\kappa$ | estimated | Relative transmission in the city after NPIs |
| $\nu$ | estimated | Slope of the sigmoid function |
| $\tau$ | varied (see text) | Midpoint of transmission reduction |
| N | 1,034,977 [17] | Population in Canton Bern |
| $N_H$ | 314 | Healthcare workforce |

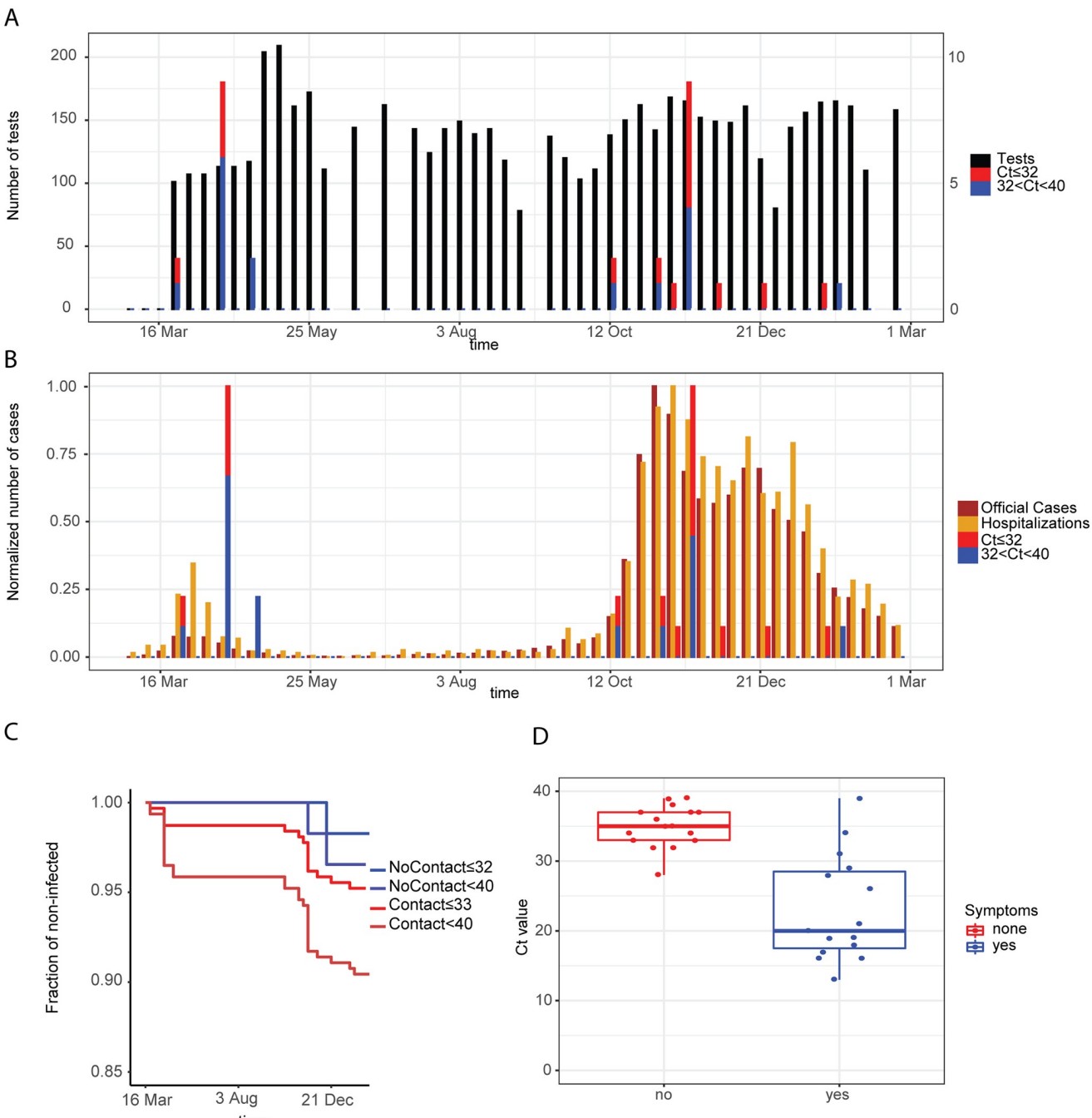

**Fig 1. Results of the testing pipeline in asymptomatic healthcare workers.** (A) Number of tests (left y-axis) and number of positive tests (right y-axis) for two different thresholds of the Ct value of the PCR test, 32 and 40, for the healthcare workers in contact with patients. (B) Number of the sum of the last 7 days of official positive cases, sum of the last 7 days of official hospitalized cases, and positive cases for the healthcare workers. (C) Kaplan-Meier curves of infected workers healthcare workers distinguishing healthcare workers and workers without contact with patients. (D) Boxplot showing the Ct value observed in the infected healthcare workers that reported having symptoms and the healthcare workers without symptoms.

The peaks of infected healthcare workers occurred with a two week delay to the peaks in the population in both waves of the COVID-19 epidemic in the Canton Bern (Fig 1B). Incidence of infections was higher in healthcare workers with patient contact compared to personnel without patient contact (Fig 1C). Low level positive results (Ct value above 32) matched with the incidence of high level positive results (Ct value below 32). High Ct values were associated with absence of symptoms (Fig 1D).

## An epidemiological model that integrates city and hospital transmission dynamics reveals an underestimation of case incidence

We have developed a transmission model that describes the infection dynamics as observed in the Swiss Canton of Bern (City Model, Fig 2A), and integrated the additional infection sources to which a group of healthcare workers might be exposed (Hospital Model, Fig 2A). In the City Model, individuals are exposed to infection at a rate $\alpha$ and become infected after a latent period. A proportion of them are hospitalised, and both hospitalised patients and infected individuals can recover and gain immunity. In the Hospital Model we additionally considered two sources of infection: infected asymptomatic coworkers (given by $\beta$) and hospitalised patients (given by $\gamma$). After infection, we considered both asymptomatic and symptomatic as infected individuals, we considered deaths only from hospitalised individuals and we did not consider reinfection. Moreover we included a variable $Q$ accounting for individuals who tested positive in the testing pipeline implemented within the department and went into quarantine and represent the fraction of detected cases before presenting symptoms.

The model was fitted to the number of hospitalised patients recorded in the Swiss Canton of Bern [17] (Fig 2B, S1 Fig), which is in line with those reported in other Swiss regions [19], as well as the infection rate in the city, $\alpha$, which is time-dependent as a results to the implemented policies. We inferred the case incidence in the region (Fig 2B), and showed that the reported data on incidence is an underestimate, likely due to official cases accounting only for test-confirmed cases. Our model predicts a true incidence $\sim$ 4.0 times higher than confirmed and reported cases during the first wave from March 1st 2020 to May 31st 2020 and up to $\sim$ 1.75 times higher from June 1st 2020 to March 1st 2021 (Fig 2B).

## Household and community transmission alone does not explain case incidence among healthcare workers

The model allowed to independently study the sources of infection that healthcare personnel might be exposed to, namely infection from the community, parameters $\alpha$ and $a$, from coworkers, $\beta$, and from hospitalized patients, $\gamma$.

In order to estimate the values of $\beta$ and $\gamma$ that can explain the actual cases detected, we fitted the detected cases to $Q$ (Quarantine variable) of our model. Of note, the fraction of detected cases corresponds to the number of pre- and asymptomatic cases detected among the workforce tested through our pipeline –i.e. the $E$, and $A$ variable in our model– and does not consider individuals who tested positive at other testing centers when already presenting symptoms (i.e. the $I$ variable in the model).

The fit was performed for the two waves of the COVID-19 pandemic separately (Fig 2C and 2D). The results reveal that in the first wave relatively high values of $\beta$ or $\gamma$ are necessary to explain our testing data best (Fig 2C left panel). Fig 2C right panel shows the fit using the values of $\gamma$ and $\beta$ that result into the best fit obtained from 2C left panel. In the second wave, the low incidence detected in our pipeline could be explained with low values of $\beta$ and $\gamma$ (Fig 2D). The best fit is shown in 2D right panel. Transmission by healthcare workers therefore seemed to be lower in the second compared to the first wave.

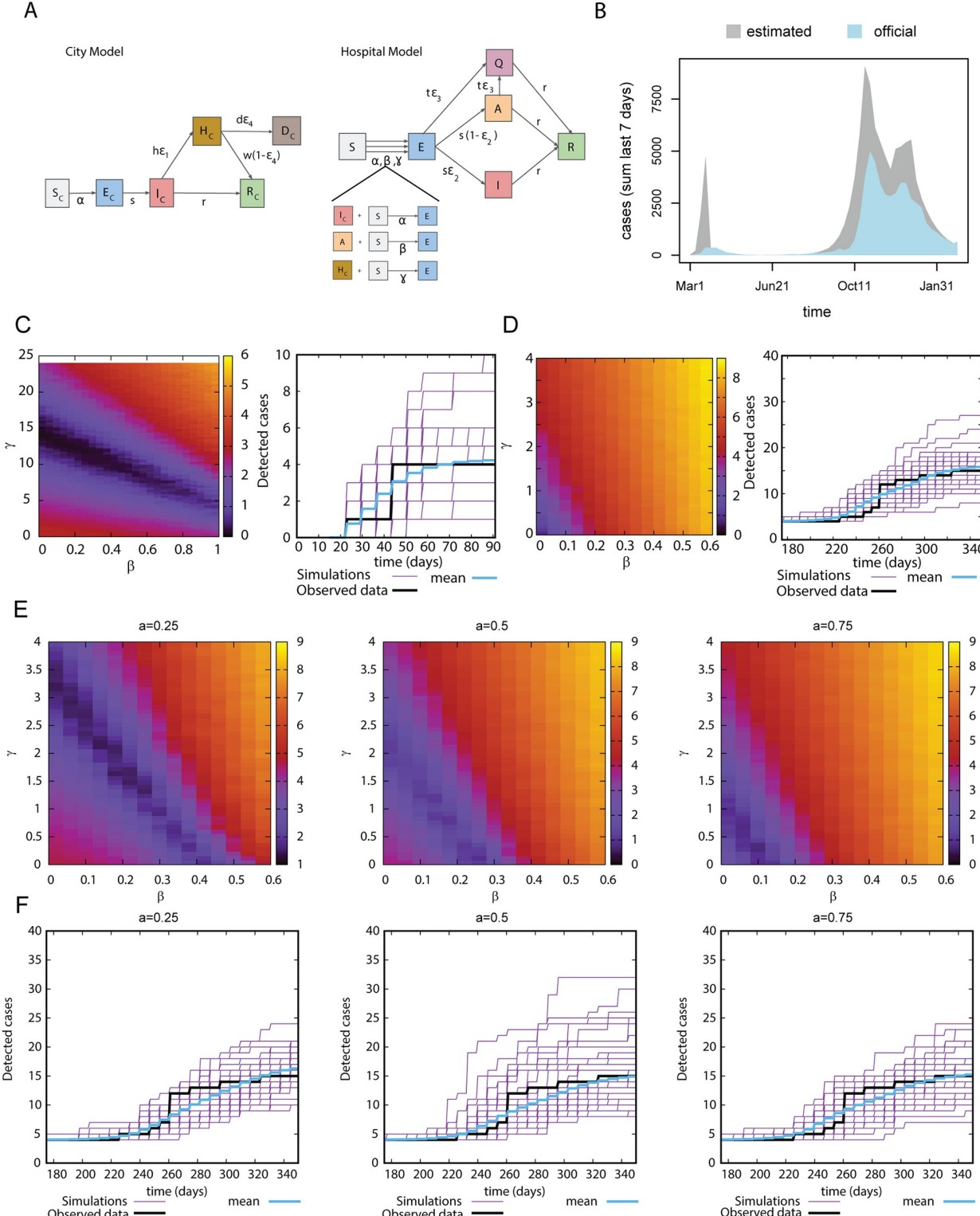

**Fig 2. Model calibration and integration of test results.** (A) Sketch of the multi-scale epidemiological model accounting for the dynamics in the city and in the hospital. (B) Number of official cases in the canton of Bern compared with the predicted new cases shown by our model. (C) left panel: Error function (see Methods) of the fit for wave 1, right panel: Observed (black curves) and simulated (50 Gillespie simulations, purple curves), and average of 1000 simulations of the 1st wave (blue curves). $\gamma = 14.72$, $\beta = 0$, $a = 1$ (D) Error function of the fit for second wave displayed as in (C). $\gamma = 0$, $\beta = 0$, $a = 1$. (E) Error function of the fit for second wave for different values of $a$. (F) Observed and simulated (50 Gillespie

simulations), and average of 1000 simulations for the second wave for different values of *a*. (left) $\gamma = 1.04$, $\beta = 0.28$, (middle) $\gamma = 0.32$, $\beta = 0.24$, (right) $\gamma = 0.08$, $\beta = 0.12$.

We tested if the protection from infection in healthcare workers outside the hospital is different compared to the general population and therefore would impact the results. Thus, we decided to study reduced infections from the community by reducing *a*. In Fig 2E, we observe that for a smaller values of *a*, an increase in $\beta$ and $\gamma$ is necessary to explain the infection rate of healthcare workers and that with decreasing *a*, the stochastic effects can become more important as we observed for high values of $\beta$. The best fits are shown in Fig 2F.

Therefore, modelling based on the results of the continuous testing of healthcare workers allowed to determine the added risks of infection at work for the two waves, and we inferred the transmission rates at the hospital that explain the observed infections.

## Regular testing and desynchronisation prevent infection transmission between coworkers

We modelled and studied two intervention measures to reduce infection transmission between coworkers: regularly testing a fraction of individuals, and desynchronisation, where the workers are split in two teams [8].

Fig 3 shows the number of infected workers as a function of *a* (Fig 3A), $\beta$ (Fig 3B) and $\gamma$ (Fig 3C) for different strategies, namely none, testing, desynchronisation and testing +desynchronisation.

The results from stochastic simulation on the effect of infection from the city, *a*, for fixed values of $\gamma$ and $\beta$, are shown in Fig 3A. As expected, the number of total infected workers increase with *a*. However, we observe no remarkable differences in the behavior of the strategies. Similarly, the number of total infected workers also increases with $\gamma$ (Fig 3C).

In Fig 3B we studied the effect of infection from co-workers $\beta$. In the case of no infection from co-workers ($\beta = 0$), testing has no impact, while desynchronisation reduces the number of cases, as a result from decreased infections from hospitalized persons. As $\beta$ increases, the number of cases and the protective effects of the measures increases. Because stochastic effects become important due to the small number of healthcare workers, the variability increases in parallel. With very large values of $\beta$, e.g. 0.6 this increase is reduced because herd immunity may be achieved.

In Fig 3D we show the effect of the frequency of testing and of the fraction of healthcare workers (fraction of testing) that is being tested. Regimes that are included range from testing every 3 days to testing every 14 days. If a specific fraction of persons is not getting tested (e.g. because of logistic reasons) can be compensated by increasing the frequency of testing.

Together, our results showed that the implementation of regular testing of healthcare workers and workforce desynchronisation are effective in preventing worker-worker transmission, and that active adherence to frequent testing is crucial for its efficiency in detecting pre-symptomatic cases.

## Regular testing outperforms desynchronisation in work output and cost-effectiveness in Switzerland

Both regular testing and desynchronisation are efficient in reducing transmission between healthcare workers (Fig 3). Nevertheless, both strategies come at a cost: an impact on the overall productivity and the cost of the tests.

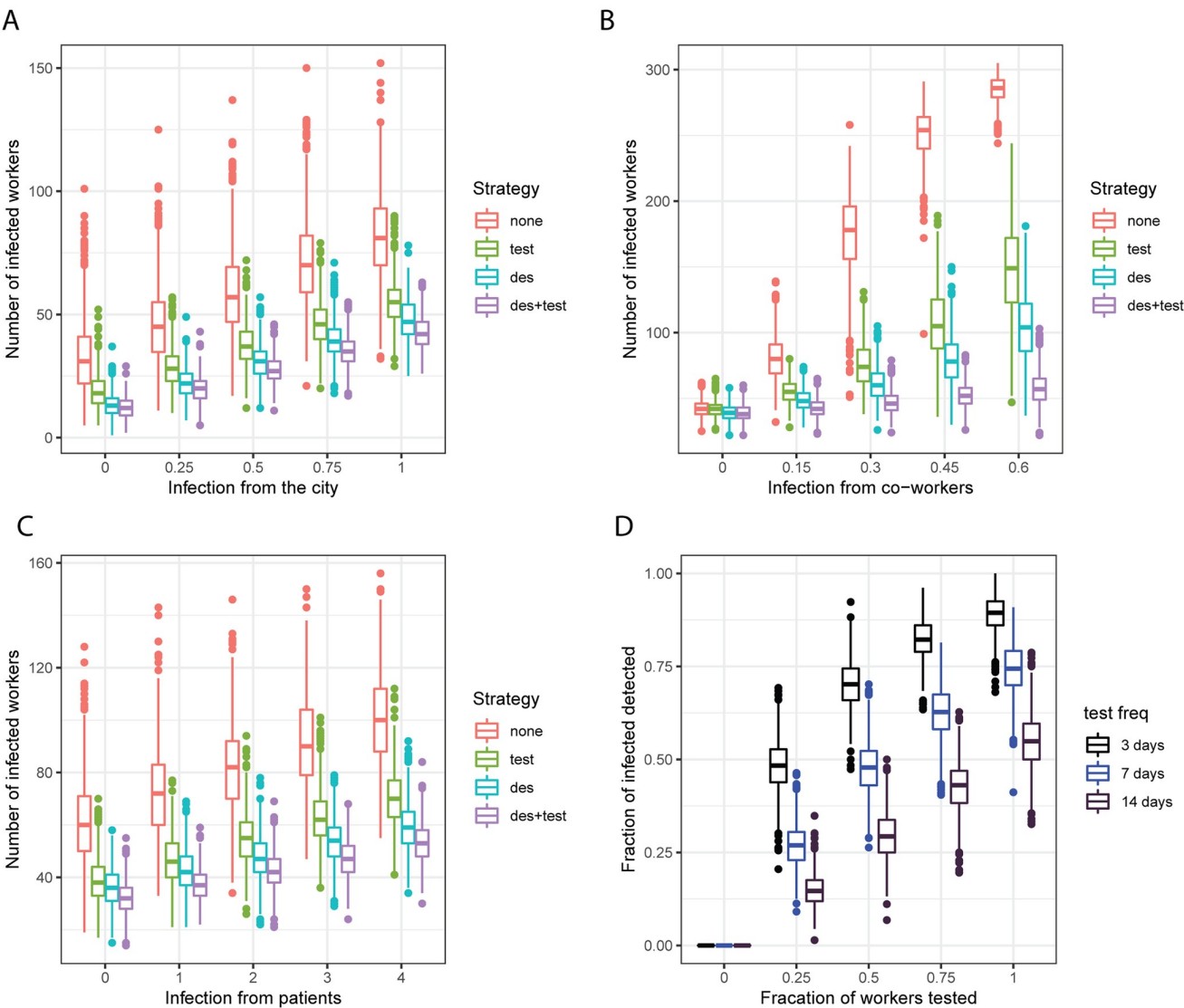

**Fig 3. Effect of the protective measures.** (A) Number of infected healthcare workers at the end of a simulation for different values of *a* for the different strategies (none, test, desynchronisation and test+desynchronisation); $\beta = 0.15$, $\gamma = 2$. (B) Number of infected healthcare workers at the end of a simulation for different values of $\beta$ for the different strategies; $\gamma = 2$, $a = 1$. (C) Number of infected healthcare workers at the end of a simulation for different values of $\gamma$ for the different strategies; $\beta = 0.15$, $a = 1$. (D) Fraction of detected cases in as a function of the fraction of workers being tested for different intervals of testing; $\beta = 0.15$, $\gamma = 2$, $a = 1$. Parameters $\beta$ and $\gamma$ are in units of days$^{-1}$. test: testing strategy; des: desynchronization strategy, des+test: combining testing and desynchronization strategy.

Most healthcare workers—notably doctors, nurses, medico-technical and laboratory staff—can carry out limited work at-home. Therefore, a notable decrease in the work output is the consequence of home-office that will be added to that of infected workers.

In order to account for work output of the healthcare cohort, we added a work variable, *W*, to the model that accounts for the work output of *S*, *E*, *A*, *R* and *Q* based on their productivity while working at-home (see Methods).

In the regular testing strategy, total work output is reduced due to workers infected and in quarantine, as well as by the time devoted to the internal testing pipeline, which we estimated as two individuals full work time per day to carry out the testing protocol. In the

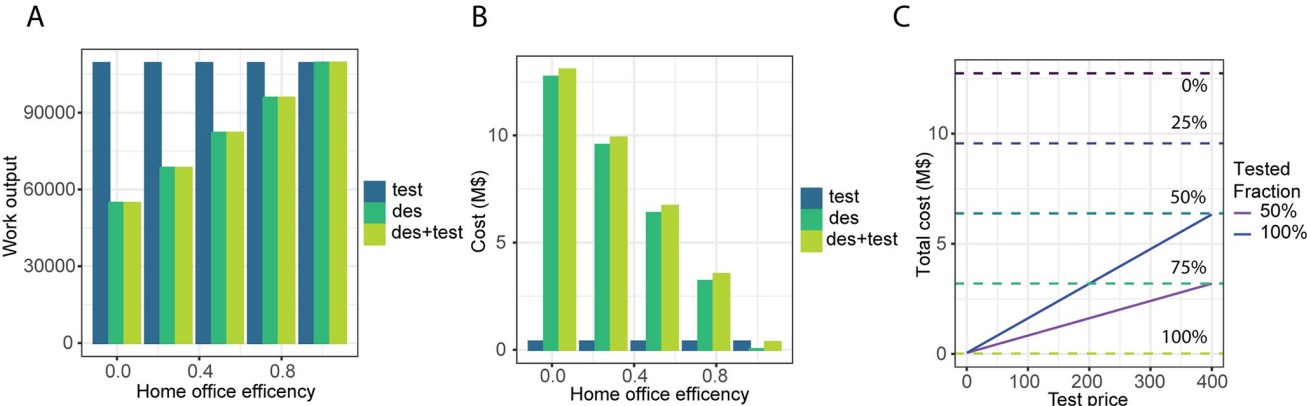

**Fig 4. Economical impact.** (A) Work output at the end of the simulation as a function of the home office efficiency for the different strategies. (B) Total cost of the measures as a function of the home office efficiency for the different strategies. (C) Total cost of the measures as a function of the cost per test. horizontal lines represent the cost of desynchronization for different levels of home-office productivity. Average of 1000 Gillespie simulations for (A, B, C). test: testing strategy; des: desynchronization strategy; des+test: combining testing and desynchronization strategy.

desynchronisation strategy, work output is reduced due to workers infected and in quarantine, as well as by the decrease in productivity during home-office.

Regular testing has a smaller impact on the overall work than desynchronisation for low home office productivity (Fig 4A). Indicating that testing would be highly efficient and recommended to reduce transmission with a minimal impact of workforce availability at different home-office productivity rates. On the other hand, weekly desynchronisation results in a stronger decrease in productivity, the magnitude of which depends on the proportion of work that is considered to be feasible at home.

Next we aim to include the cost of the test in the economic modelling. First we added the cost of the tests in US dollars (USD) and we estimated the cost of the productivity lost during desynchronisation to be proportional to the salary of the worker. For the costs of our in-house test we calculated $\sim 42$ USD per test. With a monthly average wage of 7065.20 USD, the cost of the testing is smaller than the cost of the desynchronisation, unless home office productivity approaches 100% (Fig 4B). However, if a commercially available test has to be applied then the requirements for home office productivity in the desynchronisation strategy would be lower to be cost efficient. In Fig 4C we compared the cost of testing with the cost of desynchronisation as a function of the cost per test.

In summary, our results show that for a pandemic as observed in Bern, Switzerland, a regular internal testing regime would be the optimal strategy to prevent transmission among coworkers while keeping work output high and economic cost low.

## Discussion

Protection of healthcare workers during a pandemic requires specific strategies given the elevated risks for healthcare workers from infectious agents due to their role in the treatment of infected patients. We report the results of a testing regime of asymptomatic healthcare workers during the COVID-19 pandemic. Based on the results we have developed a multi-scale epidemiological model that integrates transmission dynamics inside and outside a hospital to describe risks of SARS-CoV-2 infection for healthcare workers.

The results from the testing revealed that the peak incidence from healthcare workers lags 2 weeks behind the peak in the population. Patient contact is associated with increased incidence

of infection in healthcare workers. This is supported by other studies showing high viral titers in asymptomatic healthcare workers [20]. We showed that despite the observation that test results with high Ct-values were associated with the absence of symptoms, such results can be used to predict the prevalence of infection in healthcare workers.

Because of underreporting of official cases, especially during the first wave [11], we used hospitalization data to estimate the number of true infections in the city using publicly available data [10] that are in line with estimates in other Swiss studies [11, 19].

The results of the testing were used to build a model to determine sources of infection from inside and outside the hospital. The models show that household infection alone is insufficient to explain the rate of infections in healthcare workers.

We detected that ∼10% of healthcare workers in our cohort had been exposed (positive test with Ct value below 40) until the end of the study. This case incidence among healthcare workers is in line with that reported in other studies [3, 20]. However, estimation of the true case incidence is difficult given the seroprevalence of 1 percent healthcare workers in regions without symptomatic cases [21] up to ∼2.8% [22].

The parameters that are required to explain our observation indicate that for the first wave high infection rates from either co-workers or patients must occur. Conversely for the second wave, the model indicates a lower rate of infection from co-workers or patients. However, it remains to be determined if positive tests with low Ct values in asymptomatic healthcare workers are the consequence of infection from patients or from the community.

Next, we used our model to study how both regular testing and desynchronisation protocols can prevent infection transmission between coworkers. Both strategies are effective in reducing transmission between coworkers, in line with other studies on the benefits of desynchronisation [7–9]. Our modelling results indicate that a reduced fraction of tested persons can be compensated for by increasing the frequency of testing.

Next, we compared the consequences of the testing and desynchronisation strategies on work productivity and cost-effectiveness. Regular testing results in a minimal impact on workforce availability and work production, whereas a desynchronisation strategy would imply an important decrease on work output unless home-office productivity was virtually the same as in-hospital or in the case in which the cost of the tests is high.

Given the limited tasks that can be carried out at home by health workers—estimated to be less than 50% [14]—and the access to testing resources by medical teams, we concluded that regular testing of healthcare personnel would be the intervention of choice in Switzerland for the current pandemic. However, efforts should be made for future pandemics that prepare for solutions that allow specific tasks to be performed remotely e.g. telepharmacy.

In countries with lower income, desynchronisation would have a lower cost of implementation than testing. However, it would be necessary to asses the impact that halving the available personnel would have on the health system and on the epidemic management as a whole.

A possible limitation is that we assumed an hospitalization rate of 2.5% throughout the epidemic, from which several model parameters are then estimated. Furthermore, we did not consider reinfections among the cohort of healthcare workers over the studied period, but they could be worth including in future models over longer periods where reinfection would be more likely. Although variations in the hospitalization rate and reinfections would lead to different parameters and in turn, case incidence outside and possibly inside the hospital, these do not affect the conclusions of our study regarding the effectiveness of preventive interventions. Further limitations are that the dataset accounts for only one department and cannot be extrapolated to cases where hospitals were saturated.

In summary, our study showed that frequent and widespread testing of pre- and asymptomatic healthcare workers is effective in detecting infections and preventing transmission between coworkers while optimising work output and cost-effectiveness.

## Conclusions and outlook

We have generated a unique dataset initiated at the onset of the pandemic that allows the identification of when healthcare workers were infected outside and inside a hospital of interest. The dataset was used to calibrate a mathematical model that allowed the differentiation of the source of infections and to explore different strategies in avoiding future infections.

Future research directions may include the incorporation of other testing strategies and the impact of vaccination.

Taken together, the presented results of continuous testing of healthcare workers, and associated modelling, indicates optimal testing strategies that can be adjusted to the requirements of other institutions.

## Supporting information

**S1 File. Algorithm.** Pseudocode with the stochastic simulation algorithm.
(ZIP)

**S2 File. Data.** Data of the Internal testing pipeline.
(XLSX)

**S1 Fig. Optimization fit.** Figure legend: (A) value of $\alpha(t)$ during the simulation period. (b) sum of the last 7 days of $H_C(t)$ (red) and sum of the last 7 days of reported hospitalizations (black points).
(PNG)

**S2 Fig. Effect of $v$ constraints.** Figure legend: Effect of the constraints on $v$ on the fit (a) $\alpha(t)$ (b) Number of hospitalizations during the last 7 days (c) Number of infections during the last 7 days.
(PNG)

## Acknowledgments

## UVCM-COVID researchers

Andrew Macpherson[1,2,3,*], Daniel Candinas[1,2,3], Deborah Stroka[1,2,3], Monika Wegmüller[1], Sandra Wenger[1], Lia Bally[4], Andreas Melmer[4], Mirjam Kneubühl[4], Elke Beutler[1], Michelle Broger[1], Isabel Huber[1], Jeannine Kölliker[1], Kimberly König[1], Joseba Möri[1], Chiara Ziegler[1], Joana Freitas[4], Sophie Lagger[4], Elisabeth Leu[4], Sandra Tenisch[4], Nicole Truffer[4], Felix Alexander Baier[1,2], Patrick Brönnimann[2], Jacopo Gavini[1,2], Magdalena Eilenberg[1,2], Nicolas Melin[1,2], Daniel Rodjakovic[1,2], Annina Schmid[1,2], Riccardo Tombolini[1,2], Katharina Bacher[1,2], Marianne Berger Rentsch[1,2], Sophie Burkhalter[1,2], Marco Felber[2], Izzem Gemici[3], Dana Leuenberger[1,2], Sina Maletti[3], Sarah Maring[1,2], Jelena Murar[1,2], Philip Rubin[3], Daniela Sommer-Ezzis[1,2], Ziad Al Nabhani[1,2], Ian Young[1,2]

**1** Department of Visceral Surgery and Medicine, Bern University Hospital, University of Bern, Switzerland.

**2** Department of Biomedical Research, University of Bern, Switzerland.

**3** Bern Center for Precision Medicine, Switzerland.

**4** of Diabetes, Endocrinology, Nutritional Medicine and Metabolism, Bern University Hospital, University of Bern, Switzerland.

\*Lead author: andrew.macpherson@insel.ch

## Author Contributions

**Conceptualization:** Daniel Sanchez-Taltavull, Violeta Castelo-Szekely, Tim Rollenske, Stephanie C. Ganal-Vonarburg, Hai Li, Bahtiyar Yilmaz, Jakob Zimmermann, Michael Gerfin, Edgar Roldan, Guido Beldi.

**Data curation:** Daniel Sanchez-Taltavull, Violeta Castelo-Szekely, Shaira Murugan, Daniel Spari.

**Formal analysis:** Daniel Sanchez-Taltavull, Violeta Castelo-Szekely, Jonathan I. D. Hamley, Lilian Salm, Daniel Spari, Bahtiyar Yilmaz, Jakob Zimmermann, Edgar Roldan.

**Funding acquisition:** Daniel Sanchez-Taltavull, Stephanie C. Ganal-Vonarburg, Michael Gerfin, Edgar Roldan, Guido Beldi.

**Investigation:** Daniel Sanchez-Taltavull, Violeta Castelo-Szekely, Shaira Murugan, Jonathan I. D. Hamley, Stephanie C. Ganal-Vonarburg, Isabel Büchi, Adrian Keogh, Hai Li, Lilian Salm, Daniel Spari, Bahtiyar Yilmaz.

**Methodology:** Daniel Sanchez-Taltavull, Violeta Castelo-Szekely, Tim Rollenske, Stephanie C. Ganal-Vonarburg, Isabel Büchi, Adrian Keogh, Hai Li, Bahtiyar Yilmaz, Jakob Zimmermann, Edgar Roldan, Guido Beldi.

**Project administration:** Stephanie C. Ganal-Vonarburg, Michael Gerfin, Guido Beldi.

**Resources:** Stephanie C. Ganal-Vonarburg, Guido Beldi.

**Software:** Daniel Sanchez-Taltavull, Violeta Castelo-Szekely, Jonathan I. D. Hamley.

**Supervision:** Daniel Sanchez-Taltavull, Tim Rollenske, Stephanie C. Ganal-Vonarburg, Michael Gerfin, Edgar Roldan, Guido Beldi.

**Validation:** Tim Rollenske.

**Visualization:** Daniel Sanchez-Taltavull, Violeta Castelo-Szekely, Jonathan I. D. Hamley, Edgar Roldan.

**Writing – original draft:** Daniel Sanchez-Taltavull, Violeta Castelo-Szekely, Shaira Murugan, Tim Rollenske, Stephanie C. Ganal-Vonarburg, Isabel Büchi, Adrian Keogh, Hai Li, Lilian Salm, Daniel Spari, Bahtiyar Yilmaz, Jakob Zimmermann, Michael Gerfin, Edgar Roldan, Guido Beldi.

**Writing – review & editing:** Daniel Sanchez-Taltavull, Jonathan I. D. Hamley, Guido Beldi.

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
