## [Decision Letter · Decision Letter 0]

13 Aug 2021

PONE-D-21-23206

Regular testing of asymptomatic healthcare workers identifies cost-efficient SARS-CoV-2 preventive measures

PLOS ONE

Dear Dr. SANCHEZ-TALTAVULL,

Thank you for submitting your manuscript to PLOS ONE. After careful consideration, we feel that it has merit but does not fully meet PLOS ONE’s publication criteria as it currently stands. Therefore, we invite you to submit a revised version of the manuscript that addresses the points raised during the review process.

Please describe the details of Ct values as one of the reviewers requested.

We look forward to receiving your revised manuscript.

Kind regards,

Etsuro Ito

Academic Editor

PLOS ONE

Journal Requirements:

2. Please provide additional details regarding participant consent. In the Methods section, please ensure that you have specified (1) whether consent was informed and (2) what type you obtained (for instance, written or verbal). If your study included minors, state whether you obtained consent from parents or guardians. If the need for consent was waived by the ethics committee, please include this information.

"The study was supported by the Swiss National Science Foundation with the grants

196059 (G. B., D. S.-T.) and 196641 (S. G.-V.) and by the Bern Center for Precision Medicine (G. B., D. S.-T.). "

"The study was supported by the Swiss National Science Foundation with the grants 196059 (G. B., D. S.-T.) and 196641 (S. G.-V.) and by the Bern Center for Precision Medicine (G. B., D. S.-T.).

6.One of the noted authors is a group or consortium UVCM-COVID researchers. In addition to naming the author group, please list the individual authors and affiliations within this group in the acknowledgments section of your manuscript. Please also indicate clearly a lead author for this group along with a contact email address.

Reviewers' comments:

Reviewer's Responses to Questions

**Comments to the Author**

1. Is the manuscript technically sound, and do the data support the conclusions?

Reviewer #1: Yes

Reviewer #2: Partly

2. Has the statistical analysis been performed appropriately and rigorously? 

Reviewer #1: Yes

Reviewer #2: N/A

3. Have the authors made all data underlying the findings in their manuscript fully available?

Reviewer #1: Yes

Reviewer #2: No

4. Is the manuscript presented in an intelligible fashion and written in standard English?

Reviewer #1: Yes

Reviewer #2: Yes

5. Review Comments to the Author

Reviewer #1: Review Comments

1. In the abstract section, please add appropriate key words like testing, SARS COV 2and asymptomatic

2. Add the study design you have used in the abstract section

3. Well-articulated writing style in introduction part but didn’t show the gap why the importance of conducting this study for health workers and patients apart from entire world. Please try to state the importance of conducting this research in the revised manuscript

4. In the method part, better to include sampling procedures

5. Have you checked the confounding factors during testing? Please stated it in the revised manuscript whether you consider confounding variable or not

6. Write strengths and limitation of the study in the revised manuscript

7. Include Conclusion section following discussion part and place recommendation for future investigators

General comments

-The research is novel and will be an input for scientific community after publication

- The authors followed all scientific procedures to conduct the study except some minor comments raised in the above comments

Reviewer #2: Summary:

The authors aim to investigate a set of coupled models to explain the spread of SARS-CoV-2 not only in the Swiss Canton of Bern but also within the local hospital environment.

To do so they present two coupled models. The first is a mean-field compartmental infectious disease model to describe the dynamics of the virus within the surrounding community. The second is a stochastic model (via Gillespie simulations of related ODEs) of the viral dynamics within the healthcare workers.

To parameterize these models, the cumulative infections within Bern were used. Additionally PCR data from voluntary swabs of healthcare workers were used to parameterize the Gillespie model.

Remarks:

1. Numerous times the authors refer to the cycle threshold (Ct), though this quantity is never defined nor explained anywhere in the manuscript (in fact, I do not believe the words "Cycle Threshold" appear anywhere within the manuscript). This quantity should be defined/explained and its importance emphasized within the manuscript. (Perhaps in the RNA and RT-qPCR section). Notably some intuition should be given for the importance of the size / relative ordering of these values. What does a mean Ct of 20 intuitively mean, for the symptomatic group, compared to a Ct of 35 (Fig 1D), for example.

2. The presentation of the pseudo-code on pages 7-9 should be re-considered. In particular, the pseudocode switches between being quite high-level (Select reactions according to Gillespie's algorithm; update the states accordingly) and being quite low-level (explicit for loops in C-like syntax, etc.) In particular, the authors employ a flag variable 'Bol' with a very un-informative name. The flag, presumably, is included so as to ensure that there isn't simulated "double testing" if the Delta t chosen by Gillespie's algorithm is smaller than one day. For some reason, this flag seems to be improperly employed in the t>64 case of the algorithm. (Bol is never reset to 0 suggesting a possibility of this spurious "double testing"). The authors should ensure that this error is only present in the presentation of the algorithm and not in the actual simulations.

The authors should decide if they want to provide a high-level overview here, or a low-level implementation. If the former, then point 2d) and 2e)could be re-written to be more human-readable. If the latter, then there certainly shouldn't be any implementation glitches present within the summary of the algorithm.

Finally, a minor critique, the ending condition for the algorithm is never specified. Presumably after some time point was reached? Should step 3 read: "Go to 2(a) or END if t=365"?

3. The quantity f_p is introduced (line 171 on page 6) and then never used again or explained. (In the implementation of the algorithm, the phrase "Fraction of workers tested" is employed instead, presumably this refers to f_p). It is also unclear to me *what* f_p is. Is f_p a time dependent parameter? f_p is not present in the parameter value table, perhaps f_p is calculated on a per-day basis from the PCR data, in any case this should be made clear.

4. A few typographical issues:

- In the text on lines 180-181, the authors indicate that tau, pi, nu, and kappa are fit via minimizing this cost function. However the cost function as written is only a function of kappa, nu, and pi (notably: not a function of tau, is there a reason for this? Or is there just a typo in line 180 or in the definition of G).

- the integral between lines 181 and 182 is missing the variable of integration (presumably time).

- Lines 51-52 include the sentence "We fitted the model to the recorded number of hospitalized individuals outside the hospital". Presumably the authors mean "infected individuals outside the hospital".

- On line 164 the authors say beta_1(t) = 1.5\\beta, \\gamma_1(t)=\\gamma. and then one line 165, when the desynchronized cohorts switch, they say beta_2(t)=1.5\\beta, \\gamma_2(t)=1.5\\gamma. I assume there is a typo in the definition of \\gamma_1(t) on this interval (probably meant \\gamma_1(t)=1.5\\gamma). If this isn't a typo, this discrepancy should be explained. Also, on lines 216 and 217 they define \\gamma_1=\\gamma and \\gamma_2=\\gamma and not 1.5\\gamma, in contradiction with definitions of \\gamma_i(t) earlier and in contradiction with line 166 that states "We assume an increase of 50% in beta and gamma..."

- Perhaps a nitpicky critique, but it would be good around lines 155-167 to discuss WHEN the bare values of \\gamma and \\beta are actually used. The definition of \\beta_i switching between 1.5\\beta and 0 in lines 155-167 naturally raises the question: why not just redefine \\beta as this 50% increase of the nominal value? This isn't answered until a full two-pages later in line 248 where it becomes clear that the bare value of \\beta is used within the simulations when desynchronization is not considered.

- There are overbars on some quantities in equations (9) and (10). Are these just typographical errors or is this meant to indicate something else? Notably these bars are absent in the corresponding terms in Table 1.

5. Perhaps I am missing something, but I fundamentally do not understand the point being made in lines 283-285. While I understand that without this artificial reaction there may be situations where large values of delta t are chosen by Gillespie's algorithm, I fail to understand how this relates to errors in the solution of the ODE system. My understanding of the algorithm is as follows:

- The city ODE model is integrated for 8 days. During this time the Gillespie model is not being used.

- Then the Gillespie model is initialized and chooses a reaction and a delta t. The Gillespie state vectors are updated accordingly.

- Then the city ODE model is integrated for delta t days.

- (Then testing updates, desynch turning on and off, etc.)

I assume that the large delta t values that "lead to errors in the solution of the ODE system" that the authors refer to would be introduced at this third step of my summary. I just fail to see why that is necessarily the case. delta t, the step size for the Gillespie simulation, is not necessarily the step size for the ODE solver. It is just the final time of the time-mesh used by the ODE solver. Why not define a maximum step size, h_t say, that is sufficiently small (i.e. h_t=0.01 or smaller) and integrate the city ODE over the time mesh [t_i, t_i + h_t, t_i + 2 h_t, ..., t_i + delta t]. You then will not be sacrificing accuracy in the integration of the ODE system while not introducing an artificial reaction in the Gillespie simulation.

Partially the reason this concerns me is that with the artificial reaction I am not convinced that a Gillespie simulation is actually being employed. i.e. It almost reads as just a Monte-Carlo step with a completely spurious temporal update.

6. It is not clear to me why alpha(t) switches form throughout the simulation. It is this Holling Type-2 function during waves 1 and 2 and during the winter after wave 2, but a constant function between waves 1 and 2. Is there any explanation as to why this choice was made? What happens if between waves it is treated as having the same functional form? Additionally, it would be good to see a plot of alpha(t) for all time being considered (perhaps in the SI, at the authors' discretion).

7. The fitting/optimization process is not discussed in nearly enough detail. Is it just a brute force method over the lattice of values presented in the "Model Fitting - City" section? Is some other optimization algorithm being employed instead?

8. The authors (line 202-205) discuss how in actuality testing was performed on the in-hospital cohort only (on a voluntary basis) and not on the home-office cohort. They then simulate testing on both cohorts "for simplicity". I do not find this to be an acceptable assumption (though the authors may disagree with evidence to the contrary!). Simulating testing only once a week instead of twice a week is, admittedly, unlikely to change terribly much dynamically (and hence seems to be a reasonable assumption, though I still think it would be incredibly simple to simulate this testing twice a week instead, but that is of little consequence), however it is the other assumption here that concerns me the most: why not simulate tests on only one of the cohorts? This seems like an incredibly simple addition to add to the code, as presented, (something like 'if beta_2==0, then do the tests on E1 and A1, else do the tests on E2 and A2').

Also: why simulate these tests on *everyone*? Presumably the authors know how many people were tested within a given week and could include that relatively simply here?

9. The Epi Data on lines 62-65 should be clarified. Are these daily time series or weekly time series? Are these cumulative hospitalizations, deaths, and cases or are they only new hospitalizations, deaths, and cases?

10. Paragraph on lines 286-289 should be rewritten for clarity. This is a particularly important part of the process and comprehension is harmed by the brevity. It is not clear to me what is happening in this paragraph; additional equations here would be an asset! You're redefining the cost function from Eqn. 15, so why not just present the actual functional form of the second cost function here and reduce ambiguity. What do you mean by "minimizing the function at times t=61 and 91"? Minimizing at time t=61 will give you one answer and at t=91 will give you a second, are you equally weighting these two options? Why not minimize the function over the whole time period, as is more commonly done? Or doing some kind of sum-of-squares of weekly minimization, as you did with the original cost function in lines 180-182?

6. PLOS authors have the option to publish the peer review history of their article (what does this mean?). If published, this will include your full peer review and any attached files.

Reviewer #1: **Yes: **Mohammedjud Hassen Ahmed

Reviewer #2: **Yes: **Brydon Eastman

---

## [Author Response · Author response to Decision Letter 0]

21 Sep 2021

We thank the reviewers for their comments and suggestions that we believes have substantially improved the quality of our manuscript. We have addressed all the issues raised during the review process. A detailed explanation can be find in the ResponseToReviewers.pdf file.

---

## [Decision Letter · Decision Letter 1]

4 Oct 2021

Regular testing of asymptomatic healthcare workers identifies cost-efficient SARS-CoV-2 preventive measures

PONE-D-21-23206R1

Dear Dr. SANCHEZ-TALTAVULL,

We’re pleased to inform you that your manuscript has been judged scientifically suitable for publication and will be formally accepted for publication once it meets all outstanding technical requirements.

Kind regards,

Etsuro Ito

Academic Editor

PLOS ONE

Reviewers' comments:

Reviewer's Responses to Questions

**Comments to the Author**

1. If the authors have adequately addressed your comments raised in a previous round of review and you feel that this manuscript is now acceptable for publication, you may indicate that here to bypass the “Comments to the Author” section, enter your conflict of interest statement in the “Confidential to Editor” section, and submit your "Accept" recommendation.

Reviewer #1: (No Response)

Reviewer #2: All comments have been addressed

2. Is the manuscript technically sound, and do the data support the conclusions?

Reviewer #1: (No Response)

Reviewer #2: Yes

3. Has the statistical analysis been performed appropriately and rigorously? 

Reviewer #1: (No Response)

Reviewer #2: Yes

4. Have the authors made all data underlying the findings in their manuscript fully available?

Reviewer #1: (No Response)

Reviewer #2: Yes

5. Is the manuscript presented in an intelligible fashion and written in standard English?

Reviewer #1: (No Response)

Reviewer #2: Yes

6. Review Comments to the Author

Reviewer #1: (No Response)

Reviewer #2: (No Response)

7. PLOS authors have the option to publish the peer review history of their article (what does this mean?). If published, this will include your full peer review and any attached files.

Reviewer #1: **Yes: **Mohammedjud Hassen Ahmed

Reviewer #2: **Yes: **Brydon Eastman

---

## [Editor Report · Acceptance letter]

28 Oct 2021

PONE-D-21-23206R1 

Regular testing of asymptomatic healthcare workers identifies cost-efficient SARS-CoV-2 preventive measures 

Dear Dr. SANCHEZ-TALTAVULL:

I'm pleased to inform you that your manuscript has been deemed suitable for publication in PLOS ONE. Congratulations! Your manuscript is now with our production department. 

Kind regards, 

on behalf of

Prof. Etsuro Ito 

Academic Editor

PLOS ONE